# Preparation of Electrospun Active Molecules Membrane Application to Atmospheric Free Radicals

**DOI:** 10.3390/membranes12050480

**Published:** 2022-04-29

**Authors:** Yang Yang, Guoying Wang, Xin Li, Yves Iradukunda, Fengshuo Liu, Zhiqian Li, Hongli Gao, Gaofeng Shi

**Affiliations:** 1School of Petrochemical Engineering, Lanzhou University of Technology, Lanzhou 730050, China; 18894330389@163.com (Y.Y.); lixin950422@foxmail.com (X.L.); fs19941111@163.com (F.L.); zhiqianli163@163.com (Z.L.); 18894494436@163.com (H.G.); gaofengshi_lzh@163.com (G.S.); 2Lanzhou Institute of Chemical Physics, Chinese Academy of Sciences, Lanzhou 730000, China; i.yves05@gmail.com

**Keywords:** free radical, active molecular membrane, electrospinning, fluorescence increase rate

## Abstract

Atmospheric reactive oxygen species (ROS) play a key role in the process of air pollution and oxidative damage to organisms. The analysis of ROS was carried out by the capture-derivative method. Therefore, it is necessary to prepare an effective molecular membrane to trap and detect ROS. Electrospinning membranes were prepared by combining the electrospinning technique with chrysin, baicalein, scutellarin, genistein, quercetin, and baicalin. By comparing the structures of the membranes before and after the reaction, the fluorescence enhancement characteristics of the reactive molecular membranes and the atmospheric radicals were studied. The ability of the active molecular membranes to trap atmospheric radicals was also studied. It was found that the genistein active molecular membrane had good trapping ability in four environments. The fluorescence enhancement rates in ROS, OH radical and O_3_ simulated environments were 39.32%, 7.99% and 11.92%, respectively. The fluorescence enhancement rate in atmospheric environment was 16.16%. Indeed, the sites where the atmospheric radicals react with the active molecular membranes are discussed. It is found that it is mainly related to the 5,7 phenolic hydroxyl of ring A, catechol structure and the coexistence structure of 4′ phenolic hydroxyl of ring B and 7 phenolic hydroxyl of ring A. Therefore, the genistein molecular membrane has shown great potential in its trapping ability and it is also environmentally friendly.

## 1. Introduction

ROS (reactive oxygen species) radicals are a class of chemicals with solid oxidizing properties among atmospheric pollutants. When humans and animals inhale large amounts or are exposed to a specific concentration of ROS radicals for a long time, they can cause severe damage to cell structures and cause aging or damage to the organism, which is called oxidative stress [1,2]. The health hazards caused by ROS have received much attention [3,4]. Therefore, studying the trapping and scavenging of atmospheric ROS radicals is essential for protecting human and animal health. In measuring atmospheric free radicals, Brown R A invented fluorescent NOx bursts that trap PM_2_._5_-bound ROS radicals, which can effectively trap radicals and can be used to study the relationship between PM_2_._5_ and the growth of ROS radicals [5]. Zhou J et al. [6] measured ROS using 2,7-dichlorofluorescein diacetate (DCFH-DA) combined with a horseradish peroxide (HRP) catalyst. Their result showed that the detection instrument cannot detect free radicals directly, and demonstrated that capture-derivative methods often analyze it [7]. The tools used are large and cumbersome, easily restricted by topography, poorly mobile, and unsuitable for studying atmospheric ROS radicals’ spatial and temporal distribution.

ROS radicals are diverse (hydroxyl radicals (·OH), ozone (O_3_), singlet oxygen (^1^O_2_), superoxide anion (O_2_^−^), peroxides (HO_2_·, RO_2_) [8,9,10]) and strongly oxidizing their trapping and scavenging is very difficult. In addition, ROS radicals cannot be trapped and measured by conventional methods (e.g., trapping measurement methods for measuring gases such as PM_10_, CO_2_, and CO). Different approaches for measuring ROS radicals are electron paramagnetic resonance techniques [11,12,13,14], the fluorescent probe method [15,16,17], the chemiluminescence method [18,19], and the spectrophotometric method [20,21]; these methods mainly convert the highly reactive and short-lived ROS radicals into other electrical, fluorescent and chemical signals to measure the number of radicals indirectly.

Electrospinning, as one of the most versatile methods, has piqued the attention of many scientists in recent decades for its wide range of nanofiber fabrication options, including high aspect ratio, big specific surface area, flexibility, structural abundance, and surface functionality [22]. Shengyu Chen et al., stated that the use of electrospinning technology to create biomedical scaffolds is becoming more popular. It can control different material components to prepare nanofiber meshes with controllable structures by using a spinning polymer solution under high voltage static electricity. It also prepared an electrospinning fiber membrane with an asymmetric nitric oxide (NO) release function, which can be used for the differential regulation of cell growth. The membrane allows endothelial cells (EC) to operate normally and does not increase inflammation [23]. The natural active molecules are spun onto the film in an array using electrospinning techniques, the prepared active molecular electrospinning membranes with a large specific surface area and high porosity could increase the interaction area between active molecules and free radicals. It has also improved anti-airflow impact strength, anti-contaminant air molecular interference ability, and the ability to capture free radicals [24]. Zhan Wang et al. proved that electrospinning membranes were prepared using β-cyclodextrin-embedded quercetin, and the embedding and electrospinning processes were found to not affect the antibacterial activity of quercetin [25]. In addition, hydrophilic polymers of polyacrylonitrile (PAN) and polyvinyl chloride (PVC) nanofiber mats combined with polyvinylpyrrolidone (PVP) and polyethylene glycol (PEG) were prepared by the electrospinning method, and the results showed that the fibers could effectively trap water molecules [26]. However, heterogeneous Janus membranes with anisotropic wettability and morphology were prepared by electrospinning, and they exhibited novel directional water droplet transport and had an efficient and excellent atmospheric water collection capacity. Indeed, hybrid Janus membranes have been successfully applied to retain water in the soil during plant cultivation [27].

Thus, in this research, natural active molecules were coupled with electrospinning technology to create electrospinning membranes that may be employed independently in environmental monitoring. SEM was used to examine the microstructures of electrospun membranes. In a laboratory setting, the electrospinning membrane was put to the test in a simulated free radical environment. Finally, it was used for outdoor atmospheric environment sampling, resulting in a novel approach for atmospheric free radical collection offline. The experimental flow chart is shown in Figure 1.

## 2. Materials and Methods

### 2.1. Apparatus and Reagents

KQ5200D CNC Ultrasonic Cleaner (Dongguan Kebashi Ultrasonic Equipment Co., Ltd., Guangdong, China); AL204 Electronic Balance (Mettler-Tolley Instruments, Shanghai, China); PerkinElmer-LS55 fluorescence spectrophotometer (PerkinElmer, USA) with 1 cm four-way quartz cuvette; Model TH-150A Intelligent Medium Flow Rate Airborne Total Suspended Particulate Matter Sampler (Wuhan Tianhong Instrument Co., Ltd., Wuhan, China); SS-2533 type electrospinning equipment (Beijing Yongkang Leye Technology Development Co., Beijing, China); Model 85-2 Thermostatic Magnetic Stirrer (Guohua Electric Co., Ltd., Changzhou, China); JSM-6701F type cold field emission type scanning electron microscope (Japan Electron Optical Corporation).

Polyvinylpyrrolidone (P816208-100g, Shanghai Maclean Biochemical Technology Co., Shanghai, China); Anhydrous methanol (500 mL, analytical purity, Shanghai Xingke High Purity Solvent Co., Shanghai, China) The free radical active molecules are all analytical controls purchased from Chengdu Manster Biotechnology Co., Ltd. (Chengdu, China). with 20 mg/stick specifications and HPLC ≥ 98%, and their structures are as shown in Figure 1.

Table 1 demonstrates that since it has the basic skeleton of C _(6)_ -C _(3)_ -C_6_ in its structure, the main antioxidant group is phenolic hydroxyl, which can provide active hydrogen to inactivate free radicals and form stable semiquinone radicals.

### 2.2. Preparation of Electrospinning Membrane of Active Molecules

A certain amount of active molecule and a certain amount of polyvinylpyrrolidone (PVP) were weighed and entirely dissolved in 5 mL of anhydrous methanol. The spinning solution containing active molecules was obtained by stirring for 1 h at room temperature using a thermostatic magnetic stirrer.

The prepared atmospheric ROS radicals in situ trapping membrane can trap and scavenge them and, at the same time, have high reactivity to trap the ambient atmospheric radicals and convert them into stable reaction products. Based on the experiments to explore the active molecule electrospinning film, the conditions in Appendix A were selected as the parameters for preparing the active molecule electrospinning film.

Single-factor experiments screened the following conditions as the best conditions for preparing the electrospinning membrane of active molecules, and the screening results are shown in Table 1.

### 2.3. Fluorescence Emission Spectroscopy Detection

Fluorescence emission spectra are spectra obtained by measuring the fluorescence flux of a fluorescent substance as a function of the wavelength of the emitted light. The scanning range is 200–800 nm. The scan was repeated 3 times, the excitation wavelengths of the free radical active molecules were searched for, and the results are shown in Table 2.

### 2.4. Reaction Capacity and Free Radical Concentration Calculation

Equation (1): Fluorescence enhancement rate:(1)$$R=\frac{A-A_{0}}{A_{0}}\times 100\%$$

*R*—Fluorescence enhancement rate (%)

*A*_0_—Fluorescence signal intensity before sample sampling (FL(A.U.))

*A*—Fluorescence signal intensity after sample sampling (FL(A.U.))

Equation (2): Free radical concentration:(2)$$C=\frac{F_{X}\times K\times N\times N_{A}}{F_{g}\times t\times \left(1-\partial \right)}$$

C—Free radical concentration (10^8^/cm^3^)

$F_{x}$—Sample clearance (%)

K—Correction factor (1 × 10^−8^)

N—The amount of substance in the sample (mol)

$N_{A}$—Avogadro constant (6.02 × 10^23^)

$F_{g}$—The sampler sets the intake rate (cm^3^/min)

t—Sampling time (min)

∂—The product loss rate (5%)

### 2.5. Fluorescence Detection Precision Experiment of Active Molecular Electrospinning Membrane

The fluorescence spectra of the samples sampled from the active molecular electrospinning membrane were determined, and the relative standard deviation (RSD) was calculated to test the precision of the fluorescence detection method. The active molecular membranes from different environments were dissolved in 10 mL of methanol solution, and the same amount of 7 samples were taken. Then the fluorescence intensity was measured, and the fluorescence enhancement rate was calculated. The test results are shown in Appendix A.

Stability experiments were carried out by absorbance testing of the solution after sampling the active molecule electrostatic spinning membrane every two hours and the relative standard deviation (RSD) was calculated after measuring the stability for 8 h. Each active molecular film was sampled in different test environments and the results were measured for each test environment as shown in Appendix A. The results show that the active molecular membranes are stable in nature after sampling and that the measurements are within the permissible error range.

### 2.6. Generation and Trapping of ROS Radicals

Cigarette smoke contains large amounts of oxidizing radicals [28]; its free radicals are mainly divided into two parts.One part is the solid phase free radicals present in cigarette tar, which has a long lifetime and mainly contains quinone and PAH free radicals. The other part is the gas-phase radicals present in cigarette smoke, which have a shorter lifetime and mainly include alkoxyl radicals (RO), carbon-nuclear alkyl radicals (R’), and nitrogen oxide radicals, which are primarily present in the gas phase in cigarette smoke. Therefore, the smoke from cigarette combustion is used to simulate ROS radicals. The concentration of ROS radicals simulated in this experiment was calculated by Equation (2) as 3.24 × 10^9^/cm^3^. The active molecular membrane is set up in the sampler with the same time and concentration for detection and analysis.

### 2.7. Generation and Trapping of OH Radicals

Many experts and scholars have studied the pathway of OH radical generation in a clean atmosphere. The natural source of OH radicals in the atmosphere is the UV photolysis of ozone, that is, the ozone in the troposphere absorbs UV light at wavelengths less than 336 nm and first produces excited O(^1^D), which then reacts with water molecules in the air to make OH radicals [29]. The reaction mechanism is as follows:O_3_ + hv (λ < 315 nm) →O_2_ + O(^1^D)(3)
O(^1^D) + H_2_O →2OH(4)

According to the above equation, OH radicals are simulated to be generated in the climate simulation room of this laboratory, and ozone is generated using UV lamp irradiation. It should be noted during this period that the excited state O(^1^D) only produces OH radicals through a partial reaction. The remaining O(^1^D) is then deactivated to the ground state O(^3^P) and then re-formed as ozone by the following reaction.
O(^1^D)→ O(^3^P)(M)(5)

Thus, while using UV lamp irradiation to generate ozone, the humidifier was turned on to keep the humidity in the climate simulation room at 70% to increase the concentration of the OH radicals generated. The concentration of the OH radicals simulated in this experiment was calculated by Equation (2) as 6.50 × 10^8^/cm^3^. The active molecular membrane was set up in the sampler with the same time and concentration for detection and analysis.

### 2.8. Ozone Generation and Trapping

Near-ground ozone (O_3_) is a secondary pollutant that is irritating and highly oxidizing and can cause cardiovascular and respiratory problems [30]. The hazards posed by atmospheric ozone have received much attention [31], thus the atmospheric ozone environment is simulated in a climate simulation chamber. The concentration of ozone simulated in this experiment, calculated by Equation (2), is 8.88 × 10^8^/cm^3^. The active molecular membrane was set up in the sampler with the same time and concentration for detection and analysis.

## 3. Results and Discussion

### 3.1. Scanning Electron Microscopy Characterization of Active Molecular Membranes

Structural characterization of the prepared functional molecular electrospinning film was conducted using cold field emission type scanning electron microscopy (SEM). The SEM of the active molecular film was prepared by the electrospinning method before sampling as shown in Figure 2. The microscopic morphological characteristics of the electrospinning films of chrysin, baicalein, scutellarin, genistein, quercetin, and baicalin, as shown in Figure 2a–f, respectively. As shown in Figure 2, the surface of the active molecular film prepared by electrospinning is non-aggregated and uniformly distributed into a mesh structure, which can increase the contact area between free radicals and active substances and thus increase the efficiency of their reactions. The experimental phenomenon is consistent with the literature, a lot of research has been conducted on the preparation parameters and application of the electrospinning membrane, and it has been found that the active molecular electrospinning membrane with a uniform structure can effectively capture substances [32,33]. The three-dimensional structure can carry out the trapping response in a hierarchical manner, which is of great significance to the study of atmospheric free radical trapping [34]. In addition, the active molecular film prepared by the electrospinning method is easy to sample; natural active molecules are chosen as scavengers, which do not produce secondary pollution to the environment and are easy to degrade; they have certain advantages in their strength of resistance to airflow impact as well as kinetic performance.

The scanning electron microscopy of the flavonoids after sampling the electrospinning film is shown in Figure 3.

Experimental analysis, as well as theoretical measurement, verified that electrospinning nanofibers are extremely beneficial for trapping airborne contaminants [35,36,37]. Figure 3a–f shows the cold field emission type scanning electron micrographs of the active molecular films of chrysin, baicalein, scutellarein, genistein, quercetin, and baicalin after sampling, respectively. From Figure 3, it can be seen that the array distribution of active molecules’ electrospinning membrane is uniform after sampling, the uniform fibrous structure was destroyed, the morphology changed dramatically, and the uniform linear fibers showed a flocculent and spherical form after the reaction. The morphology of the fiber was similar to that of the fiber prepared by Guoying Wang, et al. [24] after sampling. They found that the fiber effectively captured the pollutants in the air, which led to a change in the original aligned fiber structure and the formation of flocs. The fluorescence test results showed that the active molecular membrane could trap atmospheric ROS radicals and generate stable fluorescent complexes after the trapping reaction.

### 3.2. Fluorescence Detection of Active Molecular Membranes

As illustrated in Figure 4, the blank glass fiber membrane and the active molecule carrier membrane prepared from PVP showed no fluorescence response at the excitation wavelength by fluorescence detection. The calculation by Equation (1) indicates that the glass fiber membrane used in this experiment and PVP as a polymer did not interfere in the detection of the active molecule used in the investigation, and their fluorescence increase was less than 0.004%. In addition, six active molecular membranes, including chrysin, baicalein, and baicalin, prepared by the electrospinning technique, showed only a weak fluorescence response when detected by ultrasonic treatment fluorescence. Figure 4 plots the rate of fluorescence increase before sampling for different active molecular membranes. Experimentally, it can be seen that using the glass fiber membrane without the sample as a blank, the active molecular membranes without sampling all exhibited a low fluorescence increase rate, with the most obvious one being dye lignin, which was 0.27%. This phenomenon occurs because the active molecule has a C_6_-C_3_ conjugated structure. By fluorescence detection, only a weak fluorescence reaction appears at its corresponding excitation wavelength, which can form a stable fluorescent complex under certain conditions.

### 3.3. Analysis of Simulated ROS Radical Trapping in the Chamber of Activated Molecular Membrane

The active molecular electrospinning membranes, which were treated with indoor simulated ROS sampling, were subjected to fluorescence detection, and the results are shown in Figure 5. There is a significant fluorescence enhancement in the active molecular membranes after trapping high concentrations of ROS radicals. Specifically, the fluorescence increase rate of chrysin increased from 0.11% before sampling to 21.07% after selection; baicalein rose from 0.16% increase in fluorescence before sampling to 27.73% after sampling; scutellarein increased from 0.23% increase in fluorescence before sampling to 37.98% after sampling; genistein increased from a fluorescence increase of 0.27% before sampling to 39.32% after sampling; quercetin increased from 0.23% increase in fluorescence before sampling to 30.52% after sampling; baicalin rose from 0.14% increase in fluorescence before sampling to 25.04%. After comparing the fluorescence enhancement rate before and after sampling, it was found that the enormous change in fluorescence intensity was 39.05% for genistein. This indicates that the selected active molecules reacted to effectively trap the ROS radicals in the air.

The activated molecular electrospinning membranes, which were treated with indoor simulated OH sampling, were subjected to fluorescence detection, and the results are shown in Figure 6. There is a significant fluorescence enhancement of the activated molecular membranes after trapping OH radicals. Specifically, the fluorescence increase rate of chrysin increased from 0.11% before sampling to 4.67% after sampling; baicalein increased from 0.16% increase in fluorescence before sampling to 5.65% after sampling; scutellarein increased from 0.23% increase in fluorescence before sampling to 6.50% after sampling; genistein increased from 0.27% before sampling to 7.99% after sampling; quercetin increased from a fluorescence increase of 0.23% before sampling to 5.27% after sampling; baicalin increased from 0.14% to 4.93% fluorescence increase before sampling. After the comparison of the fluorescence enhancement rate before and after sampling, it was found that the most significant change in fluorescence intensity was for genistein, which was 7.72%. This indicates that the selected active molecules effectively trap the OH radicals in the air after the reaction.

The active molecular electrospun membranes treated with indoor simulated O_3_ sampling were subjected to fluorescence detection. The results are shown in Figure 7, where there is a significant fluorescence enhancement of the active molecular membranes after trapping O_3_ radicals. This is evidenced by the increase in fluorescence from 0.11% before sampling to 6.34% after sampling for chrysin; baicalein increased from 0.16% before sampling to 8.26% after sampling; scutellarein increased from 0.23% increase in fluorescence before sampling to 9.06% after sampling; genistein increased from a fluorescence increase of 0.27% before sampling to 11.92% after sampling; quercetin increased from a fluorescence increase of 0.23% before sampling to 6.07% after sampling; baicalin increased from 0.14% to 6.03% of fluorescence increase before sampling. After comparing the fluorescence enhancement rate before and after sampling, it was found that the most significant change in fluorescence intensity was 11.65% for genistein. This indicates that the selected active molecules reacted to effectively trap the O_3_ in the air.

### 3.4. Analysis of Atmospheric ROS Radical Trapping by Reactive Molecular Membrane Environment

To use this type of reactive molecular film for atmospheric free radical sampling, it must be sampled at an outfield monitoring site. Through the collation and analysis of the environmental parameters of this testing point, in the second quarter, the pollutants at this monitoring point are in a relatively stable state, with an average AQI of 67, the primary pollutants are PM_10_ and PM_2.5_, and the number of good days in the second quarter is 81, with a ratio of 89.01% of good days. The average pollutant values for the quarter are shown in Appendix A. Appendix A shows the environmental parameters and free radical concentration, as obtained by Equation (2) around the sampling site.

The active molecular films sampled by the atmospheric environment were treated and subjected to fluorescence detection, as shown in Figure 8. There is a significant fluorescence enhancement in the active molecular films after trapping the atmospheric environment free radicals. This is evidenced by the increase in fluorescence from 0.11% before sampling to 10.33% after sampling for chrysin; baicalein increased from 0.16% increase in fluorescence before sampling to 10.79% after sampling; scutellarein increased from 0.23% increase in fluorescence before sampling to 11.77% after sampling; genistein increased from a fluorescence increase of 0.27% before sampling to 16.16% after sampling; quercetin increased from a fluorescence increase of 0.23% before sampling to 11.16% after sampling; baicalin increased from 0.14% to 10.37% fluorescence increase before sampling. After comparing the fluorescence enhancement rate before and after sampling, it was found that the most considerable change in fluorescence intensity was for genistein, which was 15.89%. This shows that the active molecular membrane prepared by the electrospinning method can have a good response in the simulated environment. The stereo-structured active molecular membrane can still react with free radicals in the atmospheric environment and have a good trapping effect on environmental atmospheric free radicals.

### 3.5. The Comparison Analysis of Experimental Results

Based on this investigation, the ability of free radical active molecules to trap and scavenge free radicals was tested by preparing six different free radical active molecules’ electrospinning membranes and using them as in situ trapping membranes for atmospheric ROS radicals, which are capable of trapping atmospheric ROS radicals. The free radical active molecules selected in this paper are flavonoids found in natural plants, respectively chrysin [38,39], baicalein, [40,41], scutellarin, genistein, [42], quercetin [43,44], and baicalin. All have a wide range of pharmacological effects, have antioxidant, free radical scavenging, anti-inflammatory, and anti-tumor effects, have a specific protective effect on the immune, cardiovascular, digestive, and neurological systems, and also have an excellent scavenging effect on reactive oxygen radicals. During the reaction between the reactive molecule and the free radical, the free radical reacts with the phenolic hydroxyl group on the benzene ring in a hydrogen capture reaction and generates H_2_O and ROOH. After the phenolic hydroxyl group is dehydrogenated, it is influenced by the resonance of the C_6_-C_3_ conjugated system to create stable semi-quinone radicals. Thus, the radical chain reaction is terminated, while the electrons of n-π* leap are significantly reduced, reducing active flavonoid molecules and producing substances with fluorescence intensity. Hence, the active substances can trap and eliminate radicals. The presence of phenolic hydroxyl groups at the A-ring 7 and B-ring 4′ positions has the most excellent effect on the reaction of the active molecule with reactive oxygen radicals. At the A-ring 7 and B-ring 4′ positions, the phenolic hydroxyl groups, due to the formation of p~π conjugation extending the C_6_-C_3_ conjugation system, promote the delocalization of electrons and prompt the generation of relatively stable reactive molecules after the hydrogen abstraction reaction radical intermediates, thus enhancing their reactions with reactive oxygen species. The presence of the 4-position carbonyl group prolonged the C_6_-C_3_ conjugated structure under the same parent nucleus, thus facilitating the formation of stable free radical intermediates by the active molecules, and therefore exhibiting specific antioxidant activity. Hence, the mechanism of the reaction between the radical and the active molecular phenolic hydroxyl group [45] and the possible molecular structure of the active molecular radical are shown in Figure 9.

Through the trapping of free radicals by the active molecule membranes and analysis, it was found that all six active molecules selected in this paper exhibited a particular increase in fluorescence intensity. The flavonoids chrysin, baicalein, and scutellarein showed a gradual increase in the rate of fluorescence increase after the reaction as the number of active molecular phenolic hydroxyl groups increased from two to four, indicating that the phenolic hydroxyl groups played a role in trapping atmospheric free radicals. The rates of fluorescence increase before and after the reaction of chrysin, baicalein, and scutellarein active molecular membrane are shown in Table 3. The hydroxyl group at the seven positions of the A ring of baicalin is replaced by a glucuronide carboxylic acid group, so its ability to trap and scavenge atmospheric free radicals is intermediate between baicalein and chrysin. It is suggested that the hydroxyl group at the seven positions of the A ring of flavonoids is the influential hydroxyl group during scavenging free radicals.

By comparing the fluorescence increase rates of chrysin and baicalein active molecules before and after the membrane reaction (see Table 3), it was found that when the number of phenolic hydroxyl groups was the same for dynein and baicalein, the different parent ring structures had a significant influence on the ability of the active molecules to trap atmospheric free radicals, specifically, when the number of phenolic hydroxyl groups was the same, the ability of the active molecules of isoflavones to scavenge reactive oxygen radicals was greater than that of the active molecules of flavonoids. By comparing the fluorescence increase rate of the active molecular membrane of scutellarein and genistein, it was found that the presence of the phenolic hydroxyl group at the 4′ position was brought together with the phenolic hydroxyl group at the seven position of the A ring, which significantly increased the antioxidant activity of this structure.

By comparing the fluorescence increase rates of active molecular membranes of baicalein, scutellarein, and genistein (see Table 4), it was found that with the same parent nucleus, the B-ring 3′ and 4′-position -catechol structures could effectively scavenge hydroxyl radicals, with the 4′-position phenolic hydroxyl group being critical. When no substituent was present in the B-ring, the o-dihydroxy group on the A-ring dihydroxy group on the A-ring had the same effect [46].

The comparison of chrysin, baicalein and baicalin showed that the fluorescence enhancement rate of the baicalin active molecule after membrane reaction was between that of baicalein and chrysin, indicating that the o-phenylene dihydroxyl group at the 5,6 position of the A-ring has the same scavenging effect on atmospheric free radicals. The glucuronide carboxylic acid group replaces the hydroxyl group at the seven position of the A-ring of baicalin. Still, its active group is more concentrated than other active molecules, and this structure may also promote the reaction of baicalin with atmospheric ROS free radicals. This structure may also enable the reaction of baicalin with atmospheric ROS radicals.

## 4. Conclusions

In this paper, active molecular nanofiber membranes were successfully prepared by the electrospun membrane, which was used to simulate ROS, OH radicals, and O_3_ environmental sampling in a climate chamber. The membrane was then applied to atmospheric environmental sampling. The structure of the membrane changed greatly before and after sampling, indicating that the membrane could effectively capture atmospheric free radicals. The fluorescence spectrophotometry test showed that the fluorescence increase rates of simulated ROS, OH free radicals, O_3_ environmental sampling and atmospheric environmental sampling of genistein electrospinning membrane were 39.32%, 7.99%, 11.92%, and 16.16%, respectively. The active molecular membrane prepared by electrospinning technology demonstrated great advantages such as convenient sampling, effective capture of atmospheric environmental free radicals, straightforward analysis and treatment, and no secondary pollution. It can be widely used in gradient sampling and offline analysis of atmospheric, ecological free radicals. The combination of electrospinning technology and natural active molecules can be applied to capture free radicals in the atmospheric environment and change the types of polymers and the number and types of active molecules to prepare other antioxidant materials.

## Data Availability

The datasets analyzed during the current study are available from the corresponding author on reasonable request.

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
