# Peer review of "Preparation of Electrospun Active Molecules Membrane Application to Atmospheric Free Radicals"

_membranes, 2022, doi:10.3390/membranes12050480_

Round 1

Reviewer 1 Report

1,The meaning of the title does not match the focus of the article.

2,The descriptions of the references cited in the introduction do not summarize and summarize with their own views, but simply list them, especially the references in the third paragraph about "electrospinning nanofilms" are not obvious for this article.

3,The full abbreviation of ROS appears for the first time in lines 25-26 of the text, and this sentence should be placed in the first sentence of the first paragraph.

4,In the text [1-2], [3-4] and other citation formats are wrong, of course, there are punctuation, capitalization errors, please check sentence by sentence.

5,The subtitle should be simplified. For example, the subtitle 2.1 is too long and verbose.

6,Figure 2 Please further explain its working principle.

7,   Why is the scale bar of d in Figure 3 different from the other 5 figures? Also, please explain the differences between the 6 fiber membranes in Figure 3.

8,Please combine with Figure 4 to analyze the principle of ROS free radical capture by active molecules in detail. Please quantitatively characterize the resulting complexes.

9,Please supplement the microscopic fluorescence image for further explanation.

10,Reduce the number of subsections in the combined text. For example, 3.2.1 and 3.2.2 can be combined into the experimental part of 2.

11,The abstract and conclusion sections should be re-condensed.

12,Please refer to the structure of the paper published in the journal Membrances.

13,The English sentences of the full text need to be polished and improved.

Reviewer 2 Report

In the manuscript, nanofiber membranes containing different active molecules were prepared by electrospinning, which were used to simulate ROS, OH radical and O3 environmental sampling in climate chamber and applied to atmospheric environment sampling. The structures and fluorescence enhancement properties of electrospun nanofibers membranes with different active molecular were compared before and after the reaction with atmospheric free radicals. It is an interesting work but there are still some issues to be addressed.

  1. The grammar for title should be modified to have a better understanding.
  2. The format of the references is inconsistent. Please check the information of all the references carefully.
  3. The last paragraph of introduction should be shortened to briefly introduce the concept and novelty of this work. Most of the work could be shifted into the discussion section.
  4. Authors should provide more background on the structure, properties and applications of electrospun fibrous membranes. Please carefully read and discuss: Core effect on mechanical properties of one dimensional electrospun core-sheath composite fibers; Structural design toward functional materials by electrospinning: A review;
  5. The reviewer believes that the data in Figure 6 and Figure 7 do not conform to statistical law. In addition, for Figure 6, 7, 8 and 9, the histogram of before sampling can hardly be seen, so it is recommended to redraw the picture.
  6. One scheme to show the experimental procedure is required to help the understanding.
  7. This reviewer thinks that Figure 3c and 3d are misused, please replace the correct picture.
  8. What detection wavelengths did the authors use for fluorescence detection? It is not given in the figure of fluorescence spectrum or text.
  9. Mechanical properties are very important for the practical applications (refer: Electrospun nanofiber reinforced composites: a review). How about the mechanical performance of these samples?
  10. Can this membrane be defined a smart membrane? Please carefully read: A review of smart electrospun fibers toward textiles.
  11. In the fluorescence spectrum in Figure 6, there is a broad peak that the authors did not interpret, which would make it difficult for the reader to understand.
  12. Please use “electrospun” as adjective, such as electrospun fibers.
  13. There are still some grammar and typos. Authors should carefully check the whole manuscript.

Round 2

Reviewer 1 Report

1, the article structure problem. "3.1 Results" should be removed. Please insert the charts and formulas in "3.2 and 3.3" into "3.1.1, 3.1.2, 3.1.3 and 3.1.4" for analysis and cannot be used as two parts of the article.

2, the format of the article. References such as [1-2], [3-4], [18-19], etc. should be changed to [1,2], [3,4], [18,19], etc. Please mark (1), (2), (3) and other numbers after the formula. Please refer carefully to other articles in Membranes Magazine.

3. The language of the article. Please polish your article again, English needs to be improved.

Round 3

Reviewer 1 Report

  1. The format of the figures and tables in the text can be further optimized. It looks messy now, and some unimportant ones can be deleted.
  2.  It is suggested that "3. Results" in the text can be changed to "3. Results and discussion", and then "4. Discussion" to "3.5 The comparison between...".
  3. The sentences of the full text also need to be refined. Please check the meaning of the sentences on lines 377 to 379. Of course also check the meaning of other sentences.

Author Response

Please see the attachments

This manuscript is a resubmission of an earlier submission. The following is a list of the peer review reports and author responses from that submission.

Round 1

Reviewer 1 Report

In this paper,the electrostatic spinning technique was combined with six free radical active molecules of Chrysin, Baicalein, Scutellarin, Genistein, Quercetin, and Baicalin to prepare a  molecular nanofiber membranes, which was used to simulate ROS, OH radicals, and O3 environmental sampling in a climate chamber, and applied to atmospheric environmental sampling, all of them have good collecting ability. In addition, the location of reaction between atmospheric free radicals and active molecular membranes is also discussed. The article carries on the comprehensive rich work, but from the manuscript reflects the lack of rigorous and serious work attitude. However, there are still some issues to be addressed. A minor revision is suggested before its acceptance.

  1. The chapter labels in the manuscript are wrong, which seriously affects readers' understanding.
  2. The manuscript abstract is not refined enough, please revise it.
  3. Some of the references are outdated and of low quality, recommended modification.
  4. The expression of “electrostatic spinning” is too old. Please correct it into “Electrospinning”. In addition, more background on electrospinning and electrospun fibers should be provided with supporting review and research papers: Structural design toward functional materials by electrospinning: A review; Electrospun fiber membrane with asymmetric NO release for the differential regulation of cell growth; Polymer Chemistry 9, 2685-2720, 2018; etc.
  5. Please adjust the two figures in Figure 5. It is suggested to describe the two figures separately, with the second one trying to make the contrast more obvious. Similarly, there are also figures 6, 7 and 8 that need to be adjusted.
  6. The scale bar of all SEM images should be rebuilt to have a better readability.
  7. All the data and data in the figures should be expressed in a scientific way with addition of standard deviation.
  8. Several times in the text” Error! Reference source not found”, please correct them. In addition, please according to the style of this journal to list the citations in the manuscript with serial numbers.
  9. Please carefully check the whole manuscript, there are some spelling and grammar problems in the manuscript. English editing service should be performed before submission.

Author Response

Reviewer 1

  1. The chapter labels in the manuscript are wrong, which seriously affects readers' understanding.

Thank you for the crucial question; we have re-edited the chapter labels of the article as follows.

The order and label of the re-edited article are:

  1. Introduction
  2. Experimental methods

2.1. Experimental instruments and equipment and experimental reagents

2.2. Preparation of electrospinning membrane of active molecules

2.2.1. Spinning solution preparation

2.2.2. Preparation of electrospinning membrane of active molecules

2.3. Reaction capacity and free radical concentration calculation

2.4. Fluorescence emission spectroscopy detection

2.5. Generation and trapping of ROS radicals

2.6. Generation and trapping of OH radicals

2.7. Ozone generation and trapping

  1. Results and Analysis

3.1 Scanning electron microscopy characterization of active molecular membranes

3.2 Fluorescence detection precision experiment of active molecular electrospinning membrane

3.2.1 precision experiment

3.2.2 Stability experiments

3.3 Fluorescence detection of active molecular membranes

3.4 Analysis of simulated ROS radical trapping in the chamber of activated molecular membrane

3.5 Analysis of simulated OH radical trapping in the chamber of activated molecular membrane

3.6 Indoor simulated O3 capture analysis by activated molecular membranes

3.7. Analysis of atmospheric ROS radical trapping by reactive molecular membrane environment

  1. Conclusion and outlook

The order of figures in the manuscript is:

Figure 1. Reaction mechanism between free radicals and active molecular phenolic hydroxyl groups and possible molecular structure of active molecular radicals.

Figure 2. Chemical structure formula of the active molecule.

Figure 3. Scanning electron micrographs of flavonoids before sampling electrospun membranes (a: Chrysin, b: Baicalein, c: Scutellarin, d: Genistein, e: Quercetin, f: Baicalin).

Figure 4. Scanning electron microscopy of flavonoids after sampling the electro-static spinning electrospinning film (a: Chrysin, b: Baicalein, c: Scutellarein, d: Genistein, e: Querce-tin, f: Baicalin).

Figure 5. Fluorescence comparison between glass fiber membrane and PVP membrane before and after sampling

Figure 6. Fluorescence and fluorescence increase rate after simulated ROS radical reaction in the active molecular membrane trap chamber

Figure 7. Fluorescence growth rate after simulated OH radical reaction in the active molecule electrospinning membrane trapping chamber

Figure 8. Fluorescence diagram after simulated O3 reaction in the active molecule electrospinning membrane trapping chamber.

Figure 9. Fluorescence increases rate over time

The order of tables in the manuscript is:

Table 1. Electrospinning parameter settings

Table 2. Table of optimal conditions for preparing an electrospinning film of active molecules

Table 3. Emission wavelengths for fluorescence detection of active molecules

Table 4. Experimental results of fluorescence detection precision

Table 5. Experimental results of fluorescence detection stability

Table 6. Pollutant averages for the second quarter of 2021.

Table 7. Fluorescence increases (%) before and after membrane reaction of active molecules of baicalein, baicalein, and wild baicalein.

Table 8. Fluorescence increase (%) of Baicalein, Scutellarein, and Genistein active molecules before and after membrane reaction.

  1. The manuscript abstract is not refined enough; please revise it.

We thank you first for your interesting comment, and we have modified the abstract like this:

Abstract: Atmospheric reactive oxygen radicals play a crucial role in air pollution and oxidative damage to biological organisms. A capture-derivative approach often performs their analysis, so preparing effective membranes for trapping and detecting reactive molecules of atmospheric radicals is necessary. In this paper, the electrospinning technique was combined with six free radical active molecules of chrysin, baicalein, scutellarin, genistein, quercetin, and baicalin to prepare an electrospinning membrane. The ability of different active molecular membranes to react to atmospheric free radicals was investigated by comparing the membrane structure before and after the reaction and the fluorescence enhancement characteristics after the reaction between the active molecular membranes and atmospheric free radicals. The genistein molecular membranes had good trapping ability in all four environments. The fluorescence increase was 39.32%, 7.99%, and 11.92% in the ROS, OH radicals, and O3 simulated environments, respectively, and 16.16% in the atmospheric environment. Therefore, the fluorescence analysis method for detecting active molecular electrospinning membrane and atmospheric free radicals is sensitive and simple, which can be effectively used to detect environmental atmospheric free radicals.

  1. Some of the references are outdated and low-quality, recommended modification.

Thank you so much for making an exciting comment; we have changed the old references to updated ones.

  • VENKATACHARI, P.; HOPKE, P. K.; GROVER, B. D.;EATOUGH, D. J., Measurement of Particle-Bound Reactive Oxygen Species in Rubidoux Aerosols. Journal of atmospheric chemistry 2005, 50 (1), 49-58.
  • Berglund, G. I.; Carlsson, G. H.; Smith, A. T.; Szöke, H.; Henriksen, A.; Hajdu, J., The catalytic pathway of horseradish peroxidase at high resolution. Nature: International weekly journal of science 2002, 417 (6887).
  • Venkatachari, P.; Hopke, P. K., Development and Laboratory Testing of an Automated Monitor for the Measurement of Atmospheric Particle-Bound Reactive Oxygen Species (ROS). Aerosol Science and Technology: The Journal of the American Association for Aerosol Research 2008, 42 (8), 629-635.
  • Kariola, T.; Brader, G.; Li, J.; Palva, E. T., Chlorophyllase 1, a Damage Control Enzyme, Affects the Balance between Defense Pathways in Plants. 2006.
  • Flicker, T. M.; Green, S. A., Comparison of gas-phase free-radical populations in tobacco smoke and model systems by HPLC. 2005.
  • Bartalis, J.; Chan, W. G.; Wooten, J. B., A New Look at Radicals in Cigarette Smoke. Analytical Chemistry 2007, 79 (13), 5103.
  • Bartalis, J.; Zhao, Y.-L.; Flora, J. W.; Paine, J. B.; Wooten, J. B., Carbon-Centered Radicals in Cigarette Smoke: Acyl and Alkylaminocarbonyl Radicals. Analytical chemistry 2009, 81 (2), 631-641.
  • Wennberg, P. O., Radicals follow the Sun. Nature 2006, 442 (7099), 145-146.
  • Jin; Bing; Su; Man-Nung; Lin; Jim; Jr-Min, Does Ozone–WaterComplex Produce AdditionalOH Radicals in the Atmosphere? Journal of Physical Chemistry A 2012.
  • Zoran, S.; Marković; Svetlana; Marković; Jasmina,; Dimitrić; Marković; Dejan, Structure, and reactivity of baicalein radical cation. International Journal of Quantum Chemistry 2012, 112 (7/8), 2009-2017.
  • Bocho?áková, H.; Paulová, H.; Slanina, J. í.; Musil, P.; Táborská, E., Main flavonoids in the root of Scutellaria baicalensis cultivated in Europe and their comparative antiradical properties. Phytotherapy Research 2003, 17 (6), 640-644.

  1. The expression of "electrostatic spinning" is too old. Please correct it into "Electrospinning." In addition, more background on electrospinning and electrospun fibers should be provided with supporting review and research papers: Structural design toward functional materials by electrospinning: A review; Electrospun fiber membrane with asymmetric NO release for the differential regulation of cell growth; Polymer Chemistry 9, 2685-2720, 2018; etc. 4

Thank you so much for the tremendous job you did in giving us fruitful comments; we have modified ' electrostatic spinning ' to ' Electrospinning ' and added research background on electrospinning and electrospinning fibers in the background section. The specific modifications are as follows:

In a study on the preparation of nanofilms using electrospinning, Xiuling Yang, etc.[22] reviewed the electrospinning nanofibers with a high aspect ratio, large specific surface area, flexibility, rich structure, and surface functionalization, which can be widely used in catalysis, drug release, batteries, and supercapacitors. Shengyu Chen, etc.[23] prepared an electrospun fiber membrane with asymmetric NO release function, which can be used for differential regulation of cell growth. The membrane retains the normal function of EC and does not aggravate inflammation. Guoying Wang, etc. [24], load the natural active molecules onto the spin film in an array using electrospinning techniques. The prepared active molecular electrospun membranes with a large specific surface area and high porosity could increase the interaction between active molecules and free radicals. it also has improved anti-airflow impact strength, anti-contaminant air molecular interference ability, and the ability to capture free radicals. zhanWang, etc.[25], electrospinning membranes were prepared using β-cyclodextrin-embedded quercetin, and the embedding and electrospinning processes were found not to affect the antibacterial activity of quercetin. ALAMIR etc. [26]hydrophilic polymers of polyacrylonitrile (PAN) and polyvinyl chloride (PVC) nanofiber mats combined with polyvinylpyrrolidone (PVP), polyethylene glycol (PEG) were prepared by the electrospinning method, and the results showed that the fibers could effectively trap water molecules. Ren Baona etc. [27] heterogeneous Janus membranes with anisotropic wettability and morphology were prepared by electrospinning, which exhibits novel directional water droplet transport and has efficient and excellent atmospheric water collection capacity. In addition, hybrid Janus membranes have been successfully applied to retain water in the soil during plant cultivation.

  1. Please adjust the two figures in Figure 5. It is suggested to describe the two figures separately, with the second one trying to make the contrast more obvious. Similarly, figures 6, 7, and 8 also need to be adjusted.

According to the needs of the article's content, we adjusted the structure of the article and added some experiments. Therefore, the order of Figure and Table is changed as follows:

The order of figures in the manuscript is:

Figure 1. Reaction mechanism between free radicals and active molecular phenolic hydroxyl groups and possible molecular structure of active molecular radicals.

Figure 2. Chemical structure formula of the active molecule.

Figure 3. Scanning electron micrographs of flavonoids before sampling electrospun membranes (a: Chrysin, b: Baicalein, c: Scutellarin, d: Genistein, e: Quercetin, f: Baicalin).

Figure 4. Scanning electron microscopy of flavonoids after sampling the electro-static spinning electrospinning film (a: Chrysin, b: Baicalein, c: Scutellarein, d: Genistein, e: Querce-tin, f: Baicalin).

Figure 5. Fluorescence comparison between glass fiber membrane and PVP membrane before and after sampling

Figure 6. Fluorescence and fluorescence increase rate after simulated ROS radical reaction in the active molecular membrane trap chamber

Figure 7. Fluorescence growth rate after simulated OH radical reaction in the active molecule electrospinning membrane trapping chamber

Figure 8. Fluorescence diagram after simulated O3 reaction in the active molecule electrospinning membrane trapping chamber.

Figure 9. Fluorescence increases rate over time

The order of tables in the manuscript is:

Table 1. Electrospinning parameter settings

Table 2. Table of optimal conditions for preparing an electrospinning film of active molecules

Table 3. Emission wavelengths for fluorescence detection of active molecules

Table 4. Experimental results of fluorescence detection precision

Table 5. Experimental results of fluorescence detection stability

Table 6. Pollutant averages for the second quarter of 2021.

Table 7. Fluorescence increases (%) before and after membrane reaction of active molecules of baicalein, baicalein, and wild baicalein.

Table 8. Fluorescence increase (%) of Baicalein, Scutellarein, and Genistein active molecules before and after membrane reaction.

  1. The scale bar of all SEM images should be rebuilt to better readability.

Thank you so much for the comment; we have adjusted the SEM image in the article and enlarged it in the same proportion to make its scale clearer and more readable.

  1. All the data and data in the figures should be expressed scientifically with the addition of standard deviation.

Thank you so much. We have been standardized the numbers and data in the manuscript. They were expressed scientifically, explicitly increasing the error display in the histogram, increasing the precision and stability experiments, and calculating the RSD of various experiments.

  1. Several times in the text" Error! Reference source not found", please correct them. In addition, please, according to the style of this journal, list the citations in the manuscript with serial numbers.

Thank you, Sorry, we have modified the ' Error! Reference source not found ', the format of the manuscript was changed, and the sequence number of the graph and table was revised again to have better readability. In addition, I have listed the citations in the manuscript by the serial number following the format of this issue.

  1. Please carefully check the whole manuscript; there are some spelling and grammar problems in the manuscript. English editing service should be performed before submission.

Thank you so much; we have double-examined the spelling and grammar of the manuscript and asked the native English speakers to revise and modify the manuscript to make it more readable.

-----------------------------------------

All changes are highlighted, please see attachment

Reviewer 2 Report

I don't think the article is finished yet, there are many structural and low-level errors that need to be carefully checked and revised!

  1. In the text, the error logo of "Error! Reference source not found" appears in lines 228, 236, 251, 214, 313, 315, and 349.
  2. The expression of the figure is not clear. If the expression in Figure 4 is not clear, it is recommended to use a three-dimensional figure. The comparison of Figure 5 is meaningless. It is recommended to use two figures to illustrate.
  3. The format of the table is not uniform, the units in Table 6 and Table 5 are not marked, and the significant figures in Table 1 and others are not uniform, etc.
  4. The formula is not standardized, such as the lack of units of various parameters in formula (1), the parameters of formula (2) can be substituted into the formula, and formulas (3), (4), (5) can supplement the reaction conditions.
  5. The most important thing is that "The ability of different active molecular membranes to trap atmospheric free radicals" is mentioned in the abstract of this paper, but there are few evaluation indexes to evaluate the capture ability in this paper, and the experiments of the evaluation indexes of the capture ability should be added.
  6. The structure of this paper is mixed. Part 2.1 is actually after part 2.9, which is incomprehensible. In addition, when discussing the potential of the reaction between atmospheric free radicals and active molecular membranes, the title of "2.1.5 Discussion" is wrong on the one hand, and on the other hand, it is more suitable to be placed in the previous part, and the reaction mechanism part is added.
  7. The English of the title and text still needs to be refined and improved!

Author Response

Reviewer 2

  1. In the text, the error logo of "Error! Reference source not found" appears in lines 228, 236, 251, 214, 313, 315, and 349.

Thank you so much for the comment. We are very sorry; we have modified the " Error! Reference source not found ", and the format of the manuscript was changed, and the sequence number of the graph and table was revised again so that it has better readability. In addition, I have listed the citations in the manuscript by serial number under the format of this issue.

  1. The expression of the figure is not clear. If the expression in Figure 4 is not clear, it is recommended to use a three-dimensional figure. The comparison of Figure 5 is meaningless. It is recommended to use two figures to illustrate.

We have standardized the numbers and data in the manuscript to express them scientifically, explicitly increasing the error display in the histogram, increasing the precision and stability experiments, and calculating the RSD of various experiments. I have added prominent three-dimensional graphics.

  1. The table format is not uniform, the units in Table 6 and Table 5 are not marked, and the significant figures in Table 1 and others are not uniform, etc.

We thank you so much, and we have unified the format and unit of the table. At the same time, due to the addition of experimental manuscript data, I modified the order of graphs and tables as follows:

The order of figures in the manuscript is:

Figure 1. Reaction mechanism between free radicals and active molecular phenolic hydroxyl groups and possible molecular structure of active molecular radicals.

Figure 2. Chemical structure formula of the active molecule.

Figure 3. Scanning electron micrographs of flavonoids before sampling electrospun membranes (a: Chrysin, b: Baicalein, c: Scutellarin, d: Genistein, e: Quercetin, f: Baicalin).

Figure 4. Scanning electron microscopy of flavonoids after sampling the electro-static spinningelectrospinning film (a: Chrysin, b: Baicalein, c: Scutellarein, d: Genistein, e: Querce-tin, f: Baicalin).

Figure 5. Fluorescence comparison between glass fiber membrane and PVP membrane before and after sampling

Figure 6. Fluorescence and fluorescence increase rate after simulated ROS radical reaction in the active molecular membrane trap chamber

Figure 7. Fluorescence growth rate after simulated OH radical reaction in the active molecule electrospinning membrane trapping chamber

Figure 8. Fluorescence diagram after simulated O3 reaction in the active molecule electrospinning membrane trapping chamber.

Figure 9. Fluorescence increases rate over time

The order of tables in the manuscript is :

Table 1. Electrospinning parameter settings

Table 2. Table of optimal conditions for preparing an electrospinning film of active molecules

Table 3. Emission wavelengths for fluorescence detection of active molecules

Table 4. Experimental results of fluorescence detection precision

Table 5. Experimental results of fluorescence detection stability

Table 6. Pollutant averages for the second quarter of 2021.

Table 7. Fluorescence increases (%) before and after membrane reaction of active molecules of baicalein, baicalein, and wild baicalein.

Table 8. Fluorescence increase (%) of Baicalein, Scutellarein, and Genistein active molecules before and after membrane reaction.

  1. The formula is not standardized, such as the lack of units of various parameters in formula (1), the parameters of procedure (2) can be substituted into the formula, and formulas (3), (4), (5) can supplement the reaction conditions.

We have checked the literature and books and revised the formula in the manuscript. The revision results are as follows:

Fluorescence enhancement rate:

(1)

R- Fluorescence enhancement rate (%)

A0- Fluorescence signal intensity before sample sampling (FL(A.U.))

A- Fluorescence signal intensity after sample sampling (FL(A.U.))

Free radical concentration:

(2)

—Free radical concentration (108/cm3)

—Sample clearance (%)

—Correction factor (1×10-8)

—The amount of substance in the sample (mol)

—Avogadro constant (6.02×1023)

—The sampler sets the intake rate (cm3/min)

—Sampling time (min)

—The product loss rate (5%)

  1. The most important thing is that "The ability of different active molecular membranes to trap atmospheric free radicals" is mentioned in the abstract of this paper, but there are few evaluation indexes to evaluate the capture ability in this paper, and the experiments of the evaluation indexes of the capture ability should be added.

We are very thankful for the exciting comment; we have combined with the content of the article, the summary of the manuscript was modified to:

Abstract: Atmospheric reactive oxygen radicals play a crucial role in air pollution and oxidative damage to biological organisms. A capture-derivative approach often performs their analysis, so preparing effective membranes for trapping and detecting reactive molecules of atmospheric radicals is necessary. In this paper, the electrospinning technique was combined with six free radical active molecules of chrysin, baicalein, scutellarin, genistein, quercetin, and baicalin to prepare an electrospinning membrane. The ability of different active molecular membranes to react to atmospheric free radicals was investigated by comparing the membrane structure before and after the reaction and the fluorescence enhancement characteristics after the reaction between the active molecular membranes and atmospheric free radicals. The genistein molecular membranes had good trapping ability in all four environments. The fluorescence increase was 39.32%, 7.99%, and 11.92% in the ROS, OH radicals, and O3 simulated environments, respectively, and 16.16% in the atmospheric environment. Therefore, the fluorescence analysis method for detecting active molecular electrospinning membrane and atmospheric free radicals is sensitive and simple, which can be effectively used to detect environmental atmospheric free radicals.

  1. The structure of this paper is mixed. Part 2.1 is actually after part 2.9, which is incomprehensible. In addition, when discussing the potential of the reaction between atmospheric free radicals and active molecular membranes, the title of "2.1.5 Discussion" is wrong on the one hand. On the other hand, it is more suitable to be placed in the previous part, and the reaction mechanism part is added.

We thank you so much; we re-edited the chapter labels of the article. The order and designation of the re-edited article are:

  1. Introduction
  2. Experimental methods

2.1. Experimental instruments and equipment and experimental reagents

2.2. Preparation of electrospinning membrane of active molecules

2.2.1. Spinning solution preparation

2.2.2. Preparation of electrospinning membrane of active molecules

2.3. Reaction capacity and free radical concentration calculation

2.4. Fluorescence emission spectroscopy detection

2.5. Generation and trapping of ROS radicals

2.6. Generation and trapping of OH radicals

2.7. Ozone generation and trapping

  1. Results and Analysis

3.1 Scanning electron microscopy characterization of active molecular membranes

3.2 Fluorescence detection precision experiment of active molecular electrospinning membrane

3.2.1 precision experiment

3.2.2 Stability experiments

3.3 Fluorescence detection of active molecular membranes

3.4 Analysis of simulated ROS radical trapping in the chamber of activated molecular membrane

3.5 Analysis of simulated OH radical trapping in the chamber of activated molecular membrane

3.6 Indoor simulated O3 capture analysis by activated molecular membranes

3.7. Analysis of atmospheric ROS radical trapping by reactive molecular membrane environment

  1. Conclusion and outlook

The order of figures in the manuscript is:

Figure 1. Reaction mechanism between free radicals and active molecular phenolic hydroxyl groups and possible molecular structure of active molecular radicals.

Figure 2. Chemical structure formula of the active molecule.

Figure 3. Scanning electron micrographs of flavonoids before sampling electrospun membranes (a: Chrysin, b: Baicalein, c: Scutellarin, d: Genistein, e: Quercetin, f: Baicalin).

Figure 4. Scanning electron microscopy of flavonoids after sampling the electro-static spinningelectrospinning film (a: Chrysin, b: Baicalein, c: Scutellarein, d: Genistein, e: Querce-tin, f: Baicalin).

Figure 5. Fluorescence comparison between glass fiber membrane and PVP membrane before and after sampling

Figure 6. Fluorescence and fluorescence increase rate after simulated ROS radical reaction in the active molecular membrane trap chamber

Figure 7. Fluorescence growth rate after simulated OH radical reaction in the active molecule electrospinning membrane trapping chamber

Figure 8. Fluorescence diagram after simulated O3 reaction in the active molecule electrospinning membrane trapping chamber.

Figure 9. Fluorescence increases rate over time

The order of tables in the manuscript is:

Table 1. Electrospinning parameter settings

Table 2. Table of optimal conditions for preparing an electrospinning film of active molecules

Table 3. Emission wavelengths for fluorescence detection of active molecules

Table 4. Experimental results of fluorescence detection precision

Table 5. Experimental results of fluorescence detection stability

Table 6. Pollutant averages for the second quarter of 2021.

Table 7. Fluorescence increases (%) before and after membrane reaction of active molecules of baicalein, baicalein, and wild baicalein.

Table 8. Fluorescence increase (%) of Baicalein, Scutellarein, and Genistein active molecules before and after membrane reaction.

  1. The English of the title and text still needs to be refined and improved!

We thank you so much; we once again examined the spelling and grammar of the manuscript and asked the native English speakers to revise and modify the manuscript to make it more readable.

In addition to the above issues, we also revised the following issues of the manuscript:

We modified the outdated references in the article. After checking, we have deleted the following considerations.

  • VENKATACHARI, P.; HOPKE, P. K.; GROVER, B. D.; EATOUGH, D. J., Measurement of Particle-Bound Reactive Oxygen Species in Rubidoux Aerosols. Journal of atmospheric chemistry 2005, 50 (1), 49-58.
  • Berglund, G. I.; Carlsson, G. H.; Smith, A. T.; Szöke, H.; Henriksen, A.; Hajdu, J., The catalytic pathway of horseradish peroxidase at high resolution. Nature: International weekly journal of science 2002, 417 (6887).
  • Venkatachari, P.; Hopke, P. K., Development and Laboratory Testing of an Automated Monitor for the Measurement of Atmospheric Particle-Bound Reactive Oxygen Species (ROS). Aerosol Science and Technology: The Journal of the American Association for Aerosol Research 2008, 42 (8), 629-635.
  • Kariola, T.; Brader, G.; Li, J.; Palva, E. T., Chlorophyllase 1, a Damage Control Enzyme, Affects the Balance between Defense Pathways in Plants. 2006.
  • Flicker, T. M.; Green, S. A., Comparison of gas-phase free-radical populations in tobacco smoke and model systems by HPLC. 2005.
  • Bartalis, J.; Chan, W. G.; Wooten, J. B., A New Look at Radicals in Cigarette Smoke. Analytical Chemistry 2007, 79 (13), 5103.
  • Bartalis, J.; Zhao, Y.-L.; Flora, J. W.; Paine, J. B.; Wooten, J. B., Carbon-Centered Radicals in Cigarette Smoke: Acyl and Alkylaminocarbonyl Radicals. Analytical chemistry 2009, 81 (2), 631-641.
  • Wennberg, P. O., Radicals follow the Sun. Nature 2006, 442 (7099), 145-146.
  • Jin; Bing; Su; Man-Nung; Lin; Jim; Jr-Min, Does Ozone–WaterComplex Produce AdditionalOH Radicals in the Atmosphere? Journal of Physical Chemistry A 2012.
  • Zoran, S.; Marković; Svetlana; Marković; Jasmina,; Dimitrić; Marković; Dejan, Structure, and reactivity of baicalein radical cation. International Journal of Quantum Chemistry 2012, 112 (7/8), 2009-2017.
  • Bocho?áková, H.; Paulová, H.; Slanina, J. í.; Musil, P.; Táborská, E., Main flavonoids in the root of Scutellaria baicalensis cultivated in Europe and their comparative antiradical properties. Phytotherapy Research 2003, 17 (6), 640-644.

    We thank you so much; we have modified ' electrostatic spinning ' to ' Electrospinning ' and added a research background on electrospinning and fibers in the background section. The specific modifications are as follows:

In a study on the preparation of nanofilms using electrospinning, Xiuling Yang, etc.[22] reviewed the electrospinning nanofibers with a high aspect ratio, large specific surface area, flexibility, rich structure, and surface functionalization, which can be widely used in catalysis, drug release, batteries, and supercapacitors. Shengyu Chen, etc.[23] prepared an electrospun fiber membrane with asymmetric NO release function, which can be used for differential regulation of cell growth. The membrane retains the normal function of EC and does not aggravate inflammation. Guoying Wang, etc. [24], load the natural active molecules onto the spin film in an array using electrospinning techniques; the prepared active molecular electrospun membranes with a large specific surface area and high porosity could increase the interaction area between active molecules and free radicals. It also has improved anti-airflow impact strength, anti-contaminant air molecular interference ability, and the ability to capture free radicals. Zhan Wang, etc.[25], electrospinning membranes were prepared using β-cyclodextrin-embedded quercetin, and the embedding and electrospinning processes were found not to affect the antibacterial activity of quercetin. ALAMIR etc. [26]hydrophilic polymers of polyacrylonitrile (PAN) and polyvinyl chloride (PVC) nanofiber mats combined with polyvinylpyrrolidone (PVP), polyethylene glycol (PEG) were prepared by the electrospinning method, and the results showed that the fibers could effectively trap water molecules. Ren Baona etc. [27] heterogeneous Janus membranes with anisotropic wettability and morphology were prepared by electrospinning, which exhibits novel directional water droplet transport and has efficient and excellent atmospheric water collection capacity. In addition, hybrid Janus membranes have been successfully applied to retain water in the soil during plant cultivation.

--------------------------------------------------------

All changes are highlighted, please see attachment

Reviewer 3 Report

the manuscript sounds interesting, but it has too many grammatical and editing errors, it also does not clearly show the purpose of the study to be carried out, the studies and results that they present do not have the units, the discussion must highlight the results obtained, and the conclusion must be clear about research contribution

Author Response

Reviewer 3

The manuscript sounds interesting, but it has too many grammatical and editing errors; it also does not clearly show the purpose of the study to be carried out, the studies and results that they present do not have the units, the discussion must highlight the results obtained, and the conclusion must be clear about research contribution.

We are very sorry to bring you a bad reading experience. Based on your opinions and the opinions of the other two reviewers, we made the following modifications to this paper:

We thank you so much; we re-edited the chapter labels of the article. The order and label of the re-edited article are :

  1. Introduction
  2. Experimental methods

2.1. Experimental instruments and equipment and experimental reagents

2.2. Preparation of electrospinning membrane of active molecules

2.2.1. Spinning solution preparation

2.2.2. Preparation of electrospinning membrane of active molecules

2.3. Reaction capacity and free radical concentration calculation

2.4. Fluorescence emission spectroscopy detection

2.5. Generation and trapping of ROS radicals

2.6. Generation and trapping of OH radicals

2.7. Ozone generation and trapping

  1. Results and Analysis

3.1 Scanning electron microscopy characterization of active molecular membranes

3.2 Fluorescence detection precision experiment of active molecular electrospinning membrane

3.2.1 precision experiment

3.2.2 Stability experiments

3.3 Fluorescence detection of active molecular membranes

3.4 Analysis of simulated ROS radical trapping in the chamber of activated molecular membrane

3.5 Analysis of simulated OH radical trapping in the chamber of activated molecular membrane

3.6 Indoor simulated O3 capture analysis by activated molecular membranes

3.7. Analysis of atmospheric ROS radical trapping by reactive molecular membrane environment

  1. Conclusion and outlook

The order of figures in the manuscript is:

Figure 1. Reaction mechanism between free radicals and active molecular phenolic hydroxyl groups and possible molecular structure of active molecular radicals.

Figure 2. Chemical structure formula of the active molecule.

Figure 3. Scanning electron micrographs of flavonoids before sampling electrospun membranes (a: Chrysin, b: Baicalein, c: Scutellarin, d: Genistein, e: Quercetin, f: Baicalin).

Figure 4. Scanning electron microscopy of flavonoids after sampling the electro-static spinningelectrospinning film (a: Chrysin, b: Baicalein, c: Scutellarein, d: Genistein, e: Querce-tin, f: Baicalin).

Figure 5. Fluorescence comparison between glass fiber membrane and PVP membrane before and after sampling

Figure 6. Fluorescence and fluorescence increase rate after simulated ROS radical reaction in the active molecular membrane trap chamber

Figure 7. Fluorescence growth rate after simulated OH radical reaction in the active molecule electrospinning membrane trapping chamber

Figure 8. Fluorescence diagram after simulated O3 reaction in the active molecule electrospinning membrane trapping chamber.

Figure 9. Fluorescence increases rate over time

The order of tables in the manuscript is :

Table 1. Electrospinning parameter settings

Table 2. Table of optimal conditions for preparing an electrospinning film of active molecules

Table 3. Emission wavelengths for fluorescence detection of active molecules

Table 4. Experimental results of fluorescence detection precision

Table 5. Experimental results of fluorescence detection stability

Table 6. Pollutant averages for the second quarter of 2021.

Table 7. Fluorescence increases (%) before and after membrane reaction of active molecules of baicalein, baicalein, and wild baicalein.

Table 8. Fluorescence increase (%) of Baicalein, Scutellarein, and Genistein active molecules before and after membrane reaction.

We combined with the content of the article, the summary of the manuscript was modified to :

Abstract: Atmospheric reactive oxygen radicals play a crucial role in air pollution and oxidative damage to biological organisms. A capture-derivative approach often performs their analysis, so preparing effective membranes for trapping and detecting reactive molecules of atmospheric radicals is necessary. In this paper, the electrospinning technique was combined with six free radical active molecules of chrysin, baicalein, scutellarin, genistein, quercetin, and baicalin to prepare an electrospinning membrane. The ability of different active molecular membranes to react to atmospheric free radicals was investigated by comparing the membrane structure before and after the reaction and the fluorescence enhancement characteristics after the reaction between the active molecular membranes and atmospheric free radicals. The genistein molecular membranes had good trapping ability in all four environments. The fluorescence increase was 39.32%, 7.99%, and 11.92% in the ROS, OH radicals, and O3 simulated environments, respectively, and 16.16% in the atmospheric environment. Therefore, the fluorescence analysis method for detecting active molecular electrospinning membrane and atmospheric free radicals is sensitive and simple, which can be effectively used to detect environmental atmospheric free radicals.

We modified the outdated references in the article. After checking, I deleted the following documents.

  • VENKATACHARI, P.; HOPKE, P. K.; GROVER, B. D.; EATOUGH, D. J., Measurement of Particle-Bound Reactive Oxygen Species in Rubidoux Aerosols. Journal of atmospheric chemistry 2005, 50 (1), 49-58.
  • Berglund, G. I.; Carlsson, G. H.; Smith, A. T.; Szöke, H.; Henriksen, A.; Hajdu, J., The catalytic pathway of horseradish peroxidase at high resolution. Nature: International weekly journal of science 2002, 417 (6887).
  • Venkatachari, P.; Hopke, P. K., Development and Laboratory Testing of an Automated Monitor for the Measurement of Atmospheric Particle-Bound Reactive Oxygen Species (ROS). Aerosol Science and Technology: The Journal of the American Association for Aerosol Research 2008, 42 (8), 629-635.
  • Kariola, T.; Brader, G.; Li, J.; Palva, E. T., Chlorophyllase 1, a Damage Control Enzyme, Affects the Balance between Defense Pathways in Plants. 2006.
  • Flicker, T. M.; Green, S. A., Comparison of gas-phase free-radical populations in tobacco smoke and model systems by HPLC. 2005.
  • Bartalis, J.; Chan, W. G.; Wooten, J. B., A New Look at Radicals in Cigarette Smoke. Analytical Chemistry 2007, 79 (13), 5103.
  • Bartalis, J.; Zhao, Y.-L.; Flora, J. W.; Paine, J. B.; Wooten, J. B., Carbon-Centered Radicals in Cigarette Smoke: Acyl and Alkylaminocarbonyl Radicals. Analytical chemistry 2009, 81 (2), 631-641.
  • Wennberg, P. O., Radicals follow the Sun. Nature 2006, 442 (7099), 145-146.
  • Jin; Bing; Su; Man-Nung; Lin; Jim; Jr-Min, Does Ozone–WaterComplex Produce AdditionalOH Radicals in the Atmosphere? Journal of Physical Chemistry A 2012.
  • Zoran, S.; Marković; Svetlana; Marković; Jasmina,; Dimitrić; Marković; Dejan, Structure, and reactivity of baicalein radical cation. International Journal of Quantum Chemistry 2012, 112 (7/8), 2009-2017.
  • Bocho?áková, H.; Paulová, H.; Slanina, J. í.; Musil, P.; Táborská, E., Main flavonoids in the root of Scutellaria baicalensis cultivated in Europe and their comparative antiradical properties. Phytotherapy Research 2003, 17 (6), 640-644.

In the background section, we have modified ' electrostatic spinning ' to ' Electrospinning ' and added a research background on electrospinning and electrospinning fibers. The specific modifications are as follows :

In a study on the preparation of nanofilms using electrospinning, Xiuling Yang, etc.[22] reviewed the electrospinning nanofibers with a high aspect ratio, large specific surface area, flexibility, rich structure, and surface functionalization, which can be widely used in catalysis, drug release, batteries, and supercapacitors. Shengyu Chen, etc.[23] prepared an electrospun fiber membrane with asymmetric NO release function, which can be used for differential regulation of cell growth. The membrane retains the normal function of EC and does not aggravate inflammation. Guoying Wang, etc. [24], load the natural active molecules onto the spin film in an array using electrospinning techniques; the prepared active molecular electrospun membranes with a large specific surface area and high porosity could increase the interaction area between active molecules and free radicals. it also has improved anti-airflow impact strength, anti-contaminant air molecular interference ability, and the ability to capture free radicals. zhanWang, etc.[25], electrospinning membranes were prepared using β-cyclodextrin-embedded quercetin, and the embedding and electrospinning processes were found not to affect the antibacterial activity of quercetin. ALAMIR etc. [26]hydrophilic polymers of polyacrylonitrile (PAN) and polyvinyl chloride (PVC) nanofiber mats combined with polyvinylpyrrolidone (PVP), polyethylene glycol (PEG) were prepared by the electrospinning method, and the results showed that the fibers could effectively trap water molecules. Ren Baona etc. [27] heterogeneous Janus membranes with anisotropic wettability and morphology were prepared by electrospinning, which exhibits novel directional water droplet transport and has efficient and excellent atmospheric water collection capacity. In addition, hybrid Janus membranes have been successfully applied to retain water in the soil during plant cultivation.

We adjusted the SEM image in the article and enlarged the SEM image in the same proportion to make its scale clear and readable.

We standardized the numbers and data in the manuscript to express them scientifically, explicitly increasing the error display in the histogram, increasing the precision and stability experiments, and calculating the RSD of various experiments.

We modified the ' Error! Reference source not found ', the format of the manuscript was changed, and the sequence number of the graph and table was revised again to have better readability. In addition, I have listed the citations in the manuscript by the serial number following the format of this issue.

We once again examined the spelling and grammar of the manuscript and asked the native English speakers to revise and modify the manuscript to make it more readable.

---------------------------------------------------------------------------

All changes are highlighted, please see attachment

Round 2

Reviewer 1 Report

Authors have addressed most of the issues. However, there are still many typos and grammar issues. A moderate language proof-reading should be performed before next submission.

Reviewer 2 Report

  1. The labeling of "Table1" in lines 103 to 104 in the article is wrong, because it should be Figure1.The order of the pictures after Figure 5 is all wrong, please check.
  2. What does the formula M represent between lines 186 and 187, please explain. The formula used in the text is not marked, so that the description of "Equation 1" in lines 263 to 264 is unclear.
  3. The contents of Table 4 and Table 5 need to be optimized or represented by a graph.
  4. "Error! Reference source not found." appears in lines 205, 206, 207, 233, 231, 233, 269, 304, 322, 344, 373, etc. Please check the full text and correct it.
  5. There are case errors in lines 430, 442, etc., and there are errors in the labeling format of the references in line 80.
  6. "electrospinning" appears in many places in the text. In fact, there are many places where "electrospun" can be used instead.

    7. The English sentences in the article need to be refined.

    8. There are many errors in details in the text, please check carefully. Such as "O3" in line 334, "Table1.." in line 103, and "Table" in line 123.

Reviewer 3 Report

the authors made big changes in the manuscript but forgot to discuss the results of the fibers obtained, that comment had already been sent to them, the electrospinning technique is very versatile and there is a lot of literature to consult about the parameters of the process, I recommend a review so they can explain their results

Round 3

Reviewer 2 Report

  1. Abstracts and conclusions need to be refined.
  2. The English sentences of the article still need to be polished. 

Reviewer 3 Report

 the authors did not follow my recommendations, it is necessary to relate the results of all the techniques used in the characterization of the material studied to have a strong conclusion.
The authors only made in this review a spelling correction and added this paragraph in the conclusions:
A lot of research has been done on the preparation parameters and
applications of these electrospun membranes, and these experiments have demonstrated that the prepared electrospun membranes of active molecules can function better under this structure. [41-43]
  but there is no discussion in their results of how they arrive at this conclusion